# Fibonacci Numbers between History, Semiotics, and Storytelling: The Birth of Recursive Thinking

Giuseppe Bianco [1,*], Angela Donatiello [2] and Bianca Nicchiotti [3]

1 Department of Mathematics and Computer Science, University of Palermo, 90123 Palermo, Italy
2 Department of Mathematics, University of Salerno, 84084 Fisciano, Italy; adonatiello@unisa.it
3 Faculty of Education, Free University of Bozen-Bolzano, 39100 Bolzano, Italy; bianca.nicchiotti@unibz.it
* Correspondence: giuseppe.bianco08@community.unipa.it

**Abstract:** The aim of this paper is to discuss the emergence of recursive thinking through the famous problem posed by Fibonacci regarding the growth of the rabbit population. This paper qualitatively analyzes and discusses the semiotic aspects raised by the students working with this historical source in the form of a story. From this perspective, the value of the historical problems as socio-cultural references (voices) and of the narrations as mediating factors to enhance students' learning of new mathematical concepts, such as recursion, is explored in depth. The focus lies on the pivotal role played by the students' construction of personal senses during in-group mathematical activities, in dialectics with the normative and mathematical meanings. It is highlighted that fostering environments conducive to dialogue among peers, as well as linking various shapes and contexts of knowledge, is necessary. Here, storytelling and history are regarded as fruitful resources aiding students in the gradual construction of a personal sense of mathematical concepts, including recursion.

**Keywords:** mathematics education; history of mathematics; recursion; Fibonacci numbers; computational thinking





## 1. Introduction: From the History to a Story

Usually, we assume that even ancient mathematics books, until some decades ago, resembled the Euclidean *Elements*, being full of technical notions and lacking any concrete applications or modelizations of the real word. However, the truth is the opposite; a complex philosophical system about mathematics (one of the many possible) with its underlying assumptions lies at the root of the *Elements*. This is clear because, since its birth, mathematics was used as a (human) language to explicitly communicate facts about nature. But this consistency is the exception. Flying over the centuries, we encounter Pythagoras who believed that the universe (*cosmos*) is (made of) number(s) and is ruled by invariable laws; Plato, who posited that mathematics is inherently real due its strong connection with the ideas of hyperuranion; and, finally, Galileo, who wrote the famous sentence in *The Assayer*:

> Philosophy is written in this grand book—I mean the universe—which stands continually open to our gaze, but it cannot be understood unless one first learns to comprehend the language and interpret the characters in which it is written. It is written in the language of mathematics, and its characters are triangles, circles, and other geometrical figures, without which it is humanly impossible to under-stand a single word of it; without these, one is wandering about in a dark labyrinth [1] (pp. 183–184).

This thought summarizes the trust in mathematics as a medium for human dialogue about reality. At the same time, in the quotation, the nature is represented as a book, so Galileo uses a metaphor from the artificial world to explain all the universe and the nature, which includes the human world. The nature is (as) a book made by women/men; nature is in such a way inside the woman/man of the Renaissance.

Besides this philosophical interpretation of mathematics, one of the many, we can search for a more pragmatic reconstruction of the relation between human knowledge and the mysteries of nature. If we examine the Babylonian Mathematics or the Egyptian one (in the *Rhind Papyrus*, for instance) or even the medieval tradition (*Abacus schools*), we find that mathematics was primarily a collection of practical problems. In this context, the pupils (often future accountants or administrators) were tasked with solving realistic problems, by abstracting from the specific situation and manipulating or varying the initial data, to obtain a (general) solution. In the following, we will deepen this underground path of the history of mathematics, focused on realistic *problems*, through the lenses of a famous author and his revolutionary book, arisen from the intersection of the Arabic tradition and the new rising medieval Western tradition—the *Liber Abaci* of Fibonacci.

### 1.1. Behind the Story

Leonardo Pisano (circa 1170—circa 1240/1250), commonly known as Fibonacci, was an Italian trader and mathematician who travelled around the mediterranean sea. Native of Pisa, he lived for many years in Algeria and visited Egypt, Syria, and Greece. In the broader history of mathematics, Fibonacci is a linking point, able to reconnect the Latin tradition with the Indian and Arabic one; he imported in Europe the "new" Hindu–Arabic numeral system (*Modus Indorum*), described in his book, *Liber Abaci* (1202). *Liber Abaci* is a practice-driven book—chapters 1–7 treat the foundation of arithmetic using the new numeral system; chapters 8–11 expose the applications to trading; chapter 12, quite half of the whole book, is on general problems; and, finally, chapters 13–15 involve the new algebraic techniques. In particular, Fibonacci put a problem in his book, appreciated by the mathematicians through the following centuries for many reasons. The problem from chapter XII, part seven, Section 30th, was as follows:

> A certain man had one pair of rabbits together in a certain enclosed place, and one wishes to know how many are created from the pair in one year when it is the nature of them in a single month to bear another pair, and in the second month those born to bear also [2] (p. 404).

These are the given data on the growth of rabbits. Fibonacci then presents a solution to this problem, step by step, starting from the first couple, which will give birth to another couple, from the second month; their sons will do the same, starting from the third month, and so on. The main idea can be an opportunity for students to develop problem-solving abilities, in several directions. In the following, we present the main points on which we focused during the design phase and during the activity in the classroom. But, first, we need to delve into the theoretical perspectives used to collect and analyze the data.

### 1.2. Recursion and Computational Thinking: A Competence for the New Millenium

Before deepening our approach, we should briefly outline the mathematical significance of the Fibonacci problem discussed above and why we are proposing this activity. The story of the rabbit breeder is usually seen as a smooth introduction, for its historical role and narrative structure, to recursion and recursive thinking. Indeed, the growth in the rabbit population follows a recursive pattern, whereby the rabbit population (the number of rabbit couples) at the n-th month is equal to the sum of the population of rabbits of the previous two months, as each fertile couple, older than one month (the population at the n−2th month), has generated another couple of rabbit, to be added to the population of the month before (n−1th month). This leads to the following formula: $F_n = F_{n-1} + F_{n-2}$. See Figure 1 below.

While recursion was not explicitly formalized in Fibonacci's time, in the same manner as today (e.g., using functions), the recursive pattern underlying this problem was already clear. Moreover, by embedding recursion in a story context, recursion becomes more accessible and can be understood as an emerging concept, rooted in a relatable real-world problem. Students are introduced to this recursive pattern, in comparison and contraposition with an iterative and naïve approach. Both iterative (I.T.) and recursive (R.T.) thinking are closely

related to computational thinking (C.T.). According to a famous sentence by Peter Deutsch: "to iterate is human, to recurse, divine". This synthetizes the gap between I.T. and R.T. In the following, we will briefly talk about iterative and recursive functions, without a loss of generality, because all the Computer Science interest in I.T. and R.T. moves around functions (in a Computer Science meaning), and functions can work well enough to do all that we want.

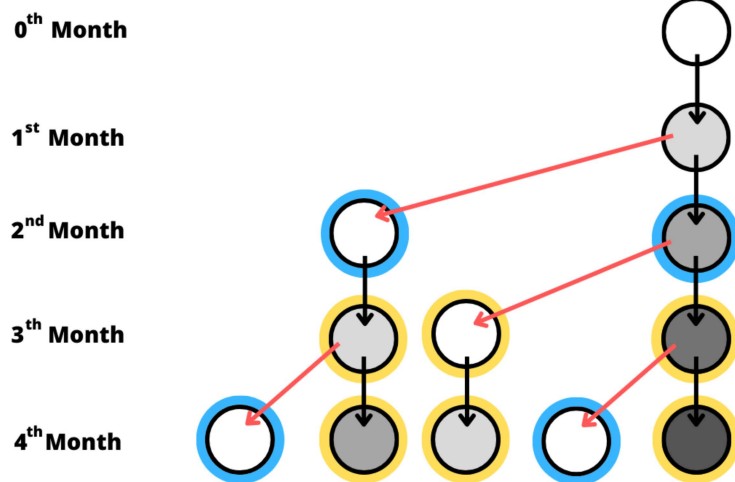

**Figure 1.** A representation of the growth of the couples of rabbits in the first months, according to the Fibonacci problem. In yellow is $F_{n-1}$, in blue is $F_{n-2}$. Inspired by [3], page 19, figure 11.

The core of iteration is the presence of a loop, in which "a sequence of steps [is repeated] as often as necessary, and appropriate repetitions of quite simple steps can solve complex problems" (Encyclopedia Britannica). The condition and number of times the sequence of steps is repeated is known from the start. This is the main difference with the recursion, more elegant, powerful, and "dangerous"; in the sequence repetition, we find the function involving itself. This definition seems circular, so an explicit rule is needed to move in the "right" direction and to give sufficient information to compute a function from the previous known steps, as well as to provide a starting point in which to find a foundation.

Using a metaphor, iteration and recursion are akin to two styles of thinking, such as writing and composing code; in such a way, if we think about code as a text, iteration and recursion are two literary genres, like a novel and a poem, one closer to daily speech and writing (prose), the other more abstract and unusual (poetry); nevertheless, both are part of the literature tradition or, outside of the metaphor, the algorithm construction. The computational cost or intelligibility changes drastically, but this is not our concern now.

Broadening our perspective, C.T., proposed sometimes with the four *Cs* (communication, critical thinking, collaboration, and creativity), is a complex process; "it includes concepts like logic, algorithms, patterns, abstraction, generalization, evaluation, and automation. It also means approaches like 'decomposing' problems into subproblems for ease in solving, creating computational artifacts (usually through coding)" [4]. The phases encompassed by C.T. are sometimes called the three *As*—"Abstraction", "Automation", and "Analysis". We can see, as such, how C.T. can be interesting from a mathematics education point of view and as a growing topic by itself; C.T. deeply intersects the more traditional interests of the mathematics education field [5]. In recent years, the discussion about coding, from primary to high school, using high-level programming languages (working on the ideas and caring less about the syntax of the coding language) has spread even beyond the scientific and educational fields, now involving the public sphere [6]. Creating, testing, and debugging algorithms is just a part of a bigger cognitive switch oriented to match the currently available and usable technologies, to enhance pupils and future citizens to work efficiently or live comfortably. Working on code or algorithms at school, we see that as C.T. involves the operative—more "concrete"—counterpart of many mathematical and

traditional problems, which can be now expressed as clear instructions, which a machine can, presently, perform in a non-ambiguous way [7].

## 2. Theoretical Framework

The theoretical framework of this paper draws from three main perspectives within mathematics education. Firstly, the literature regarding the role of history and storytelling in mathematics education is used to strengthen the background of the designed activity (Sections 2.1 and 2.3). Secondly, a semiotic perspective is employed to analyze the collected data, to highlight the dialectical and dialogical relationship between students' perspectives and the struggle to face and re-interpret the mathematical task (Section 2.2). These three perspectives are integrated in the data analysis section, as follows: history is proposed as a story (a problem extracted from a historical source) and the historical perspectives assumed highlight both the semiotic aspects [8–10] and the dialogical work on the story/source, with the aim of developing a personal view on a scientific concept [11,12], as is the case in R. T.

### 2.1. History as an Environment for Dialogue

The literature on the use of history of mathematics for educational purposes, within the field of mathematics education, is extensive (see [13], for instance). Here, we recall the perspectives we will assume below. Beyond using history and storytelling as a context to enhance motivation and give meaning to the mathematical problems and concepts, history can be useful for "looking for geneses of mathematical ideas or contexts of emergence of mathematical thinking, in the aim of defining conditions which have to be satisfied in order for the students to develop these ideas and thinking in their own minds" [14] (p. 155), on an epistemological level [15]. This does not necessarily lead to the phylogenesis–ontogenesis contention. Our perspective is more complex and is inspired by a socio-cultural perspective routed in Vygotsky and his school's theories. As stated in Boero et al.

> The child cannot be left alone to pursue this process [the development of everyday concepts] because theoretical knowledge has been *socially* constructed in the long term of *cultural history* and cannot be *re*constructed in the short term of the individual learning process. In short, 'exposure' to theoretical knowledge is necessary, and must be provided together with explicit links to children's knowledge [11] (p. 82, italics of the authors).

In this paper, the first-hand historical source serves as the cultural factor, the *voice*.

According to Radford, knowledge and its evolution (change) can be framed only within its social, historical, material, and symbolic context, considering epistemological factors as inextricably bounded and embedded in a broader sociocultural frame [8].

The activity design is inspired by Boero et al.'s "voices and echoes game" paradigm [11,12]. According to this perspective, the *voices* of "historical sources capable of conveying the crucial ideas of a scientific revolution in a concise manner" [14] (p. 155) are selected, adapted, interpreted, and presented in the classroom to initiate a dialogue, a chain of *echoes* [11]. Behind this choice, there is a clear assumption about the role (one of the many) of the history of mathematics inside the mathematics education field; the "point of departure is the fact that some verbal and non-verbal expressions (especially those produced by scientists of the past) represent in a dense way important leaps in the evolution of mathematics and science" [15] (p. 165). This is coherent to the Vygotskian perspective on the genesis of scientific knowledge, strictly connected, but not totally traceable, to everyday practice and knowledge. This gap between everyday knowledge and scientific knowledge must be filled by the interaction between teachers, experts, and students, as stated in the quoted sentence.

Both Radford's and Boero et al.'s perspectives are coherent and developed, working on the Vygotskian paradigm. Boero et al.'s frame emphasizes linguistic and historical sources as artefacts for classroom activities and supports both the background of the design of the activity and the analysis of the students' feedback and protocols [11,12]. On the other hand, Radford's lens presents a broader point of view on the whole activity and can be added in a fruitful way to a semiotic perspective, as described below.

### 2.2. A Peircian Semiotic Perspective

To track the evolution and dialectical construction of scientific concepts by students (in our case, the emergence of Recursive Thinking), a semiotic perspective serves as an initial and preliminary level of analysis. It is acknowledged that the semiotic perspective of Peirce is more flexible than the dualistic Saussurean one [16], primarily because a fixed common and interpersonal signification/meaning is not given, and it is something to build and reach collectively in the activity [9]. The model of sign by Peirce is rooted in the following three points: the *representamen* (the physical sign that stands for something), the *object* (the something denoted/pointed by the sign), and the *interpretant* (which means, not the interpretation or the interpreter, but the reaction inside the interpreter given by the sign). As Bagni in one of the most complete definitions of interpretant explains, the interpretant is:

> the reaction of the interpreter on the basis of a system of signs, culture elements, collective behaviors characteristic of a time period and social setting [17] (p. 140, translation by the authors).

Unlike the dualistic vision of Saussure (*signifier—signified*), here, the signified is split, on one hand, in the "real" object and, on the other hand, in the interpretant given by the interplay between the sign, the interpreter, her/his ideas, and the context. This does not mean that, as given, the Peircean perspective is better for educational purpose in respect to Saussure's one, but that it opens up to a plurality of vision concerning the same object and the same sign and this, in some cases, can be better than a sharp and clean definition, as granted by the latter. Moreover, the classification of Peirce becomes more subtle; the relationship between the object and its representamen leads to the following three kinds of signs: *icons*, *indexes*, and *symbols*. For iconic signs, the representamen is similar to (*resembles*) the object; for indexes, the representamen is *physically connected* with the object (e.g., smoke *means* fire); and, finally, symbols stands for the object in virtue of a *conventional norm* [17,18]. The philosophy behind this categorization is huge and, here, we do not have the time and space to deepen it. The key point to stress here is the interplay between these semiotic aspects, which often overlap, in the signs used in performing mathematics [16,18]. It is not meaningful to put each sign in fixed categories; what matters is the way in which we, as human beings, use them and this can change according to the space–time, aim, as has been previous acknowledged. This use (note we are switching to a pragmatic approach) changes according to the actors involved and the context of action. Such an open perspective naturally leads to a cross-semiotic approach, whereby the switch from a graphic to a formalized language, or passage through the natural language, can point to a change in the use/learning by the student of the signs involved; but even in the absence of this change on the syntactical level, an evolution on the semantic or on the pragmatic level can happen. We will see that the flow from natural language to the figural level, to the formal one, and again to the natural language implied a change (evolution and revolution) in students' cognition. And even in the persistence of graphic aspects, the students changed their interpretation and use of the signs involved and created during the activity.

### 2.3. Storytelling as a Setting for Meaningful Problems

In our perspective, the narrative setting serves as a landscape on which to work mathematically. However, the story, in this case, is embedded in history, lending significance to the mathematical core [19]. The Fibonacci problem is, indeed, presented in a narrative format, even if we do not consider its historical value. Furthermore, viewing a historical word problem as a story opens up a multivocal dialogue not with *the* story, but with *a* story, one of the many. This is coherent with our comprehensive approach to the history of mathematics as well, whereby past mathematical achievements are not seen as mistakes along a linear progression ("evolution"), which leads inevitably to us. Instead, history, as stories, is built by human choices and conventions, wherein their significance lies not in the necessity/destiny, but rather in the context and aim which inspired individuals who came before us, whether explicitly or implicitly.

Moreover, the narrative elements interact with the recursive pattern in a dialectic manner, while the narration is finite, recursion is potentially and humanly infinite; narration avoids repetition (except for memory help, e.g., in epics), while recursion builds on repetition with minor changes; narration looks for persuasion, recursion for performance. This does not mean intriguing connections between arts and scientific concepts are impossible; on the contrary, stories are fruitful backdrops to communicate scientific ideas to the society [20,21] or even new scientific paradigms to pupils [22–24].

## 3. Research Questions

The research problem addressed in this paper is "how can a historical mathematical problem become meaningful to present-day students?". The path chosen to deepen this idea relates to the theoretical framework. On one hand, the chosen problem is still a story, making it accessible to students as a story, detached from their daily context, yet still intelligible (for the language and the objects involved). On the other hand, the problem presents itself as a word problem and it can open the space for a semiotic (cross-semiotic) dialogue by the students to figure out the pattern briefly described in words by Fibonacci and to give meaning to this problem (indeed, a personal sense [25]). The narrative aspect is then the *fil rouge* that links the historical and the mathematical cores; the entire classroom activity centers around the comprehension, understanding, interpretation, and re-creation of the/a story and so the flow from *the* story (from history) to a (multitude of) story (born by each student, each group, or the whole classroom).

The research questions are as follows:

RQ1. Are historical and narrative settings able to create a free space for a cross-semiotic mathematical dialogue, inclusive of each student? How do students move from the given problem, through developing a personal sense, arriving at a shared meaning of the problem?

RQ2. Can storytelling bridge the gap between historically meaningful mathematics problems and everyday reality problems? Can narrative aspects assist students in reframing historical mathematical problems to match their own day-to-day mathematical experience?

## 4. Method: Design of the Activity

The activity described and the data qualitatively collected come from a two-day experience in a grade 10 classroom at a vocational high school in a major city in northeastern Italy.

The activity comprises four phases, all performed as group activities, except for the last one, as follows:

(a) in-group comprehension of the given problem and elaboration of an approach/solution;
(b) communication to the whole class of a solution;
(c) in-group construction of a new and more "clear" version of the traditional Fibonacci word problem;
(d) individual completion of a questionnaire regarding the activity.

The first step is shaped as an in-group problem solving activity (Section 4.1). The second involved a discussion with the teacher and the researcher and other groups (Section 4.2). The third step is shaped as an in-group problem posing activity (Section 4.3). Finally, the last step is a simple survey phase (Section 4.4).

### 4.1. In-Group Problem Solving

At the onset of the activity, a reshaped version of the Fibonacci problem was shown:

"A man has a pair of rabbits; these rabbits are kept together in his garden. One would like to know how many pairs of rabbits are generated from the initial pair in a year, knowing that this pair in one month gives birth to another pair, and from the second month onwards the newborns will also begin to generate new pairs".

After, a slide with the following steps was shown:

Step 1. Do you understand the problem? Do you understand what is required? Try to set up a solution strategy. As a help you can make use of drawings or sketches, simulating concretely what happens. Attempt to find a solution.

Step 2. Try to express what you have obtained in arithmetic language.

Step 3. Construct a table showing the number of pairs of rabbits for each month.

Step 4. What do you notice in the table? Can you find the number of rabbits after 13 months using the table?

These steps were designed to help students to engage with the problem, to develop a strategy, to obtain a solution, and to communicate it (phase a); this prompted students to carefully read the text multiple times, figure out the pattern for the first steps during a trial and error stage, sketch a solution using drawings, and, finally, try to express the findings in arithmetic and natural language. According to the idea of [8], we will see as students have been encouraged to make the mathematical assumption of Fibonacci explicit, meaningful to his aim and expertise and coherent to his space–time context. If knowledge is socially constructed, a Fibonacci problem should be decrypted according to his and its historical and cultural factors and should, therefore, be interpreted and adapted by students according to their own new sociocultural context. Context, in this sense, means even the daily experience of students both in school and out of school, e.g., pupils living in cities have little to no experience of breeding farms, while this can be quite a common topic for families living in the countryside. This social dimension of knowledge has been strengthened by the choice of working in groups and of communicating the strategies and solutions between groups, pushing students to create or adapt shared signs (steps 1, 2, and 3) for the new given task [9]. Moreover, the in-group and between-groups dialogue enhanced (phase b), a "hybrid semiotic matching of different views" [15] (p. 164). Speech and language are key factors in this communication-based process. Here, the choice to start from a semiotic level using the flexible approach described in Section 2.2, where the language, drawings, and numbers create an interactive and fruitful environment for a collective brainstorming.

*4.2. Classroom Discussion*

After the first in-group phase, each group selected a student to present the strategies developed by the group. These students briefly discussed their group's approaches with the classroom, the teacher, and the researcher, using a digital blackboard. As recalled in the introduction, Fibonacci represents a paradigm shift [26] in the Western tradition, transitioning from a Pythagorean and Platonic heritage to a new medieval tradition, a result of the grafting of Hellenistic sources and Arabic influences. The dialogue between the *voices*, representing shared *meaning* [25] (by the authorities—teacher, researcher, and even the indirect voice of Fibonacci), and *echoes*, representing personal *sense* [25] (by the students) is highlighted during the last phase, where students are asked to reframe and rewrite the Fibonacci problem in their own language, as a problem to be presented in another ideal classroom [11,12]. In this way, the dialectics of meanings and senses comes to a synthesis in the students' outputs.

*4.3. In-Group Problem Posing*

This third part, dialectical being the first (Section 4.1), is initiated with one suggestion and one assignment, as follows:

- How do you find the description/story by Fibonacci? Is it verisimilar? There are several implicit assumptions, which ones?
- Rewrite Fibonacci's story: work on the initial text and shape it as a story as you understood it.

The first step is auxiliary to make students aware of the hidden meanings of the problem, i.e., the mathematical viewpoint, the limited knowledge about biology of the author (due to the 8 centuries between him and us), and other modeling choices (e.g., are rabbits immortal?). This forced students to re-think the problem from their perspective,

not just as an in-school meaningless problem created by the teacher. The second point encourages active problem posing (now the problem is perceived by the students as their problem), in the form of storytelling by the students. According to what was discussed in Section 2.3., if the Fibonacci story becomes just *one* story, we can start to spread and generate new stories, similar, more clear, more funny, easier to remember, more closely connected with the mathematical core and the traditional one. Considering the author as *a voice*, can open up a series of *echoes,* raised by the voices of present-day students [12].

*4.4. Questionnaire*

Below, we report the questions present in the questionnaire:

1. How did you start tackling the problem? At what stage (and from which month) did you perceive the need to start formalising the problem using a standard mathematical language (e.g., numbers)?
2. At what stage/moment (and starting from which month) did you perceive the emergence of recursive thinking?
3. Are there contexts in everyday life where a similar reasoning structure to the one you used to arrive at the solution might be present?
4. Did working in group help you to change your point of view?
5. Did you pursue different paths within the group? If so, what were they?
6. In the end, did you reach a synthesis so that all members of the group agreed, or did you perceive that the majority forced you to accept a particular proposal?

**5. Data Analysis and Preliminary Results Discussion**

*5.1. In-Group Problem Solving*

In this section, we will analyze the protocols of the groups, on a semiotic level, to highlight the movement on the syntactical level between different representations of the problem and the dialectical relationship between the personal sense (the echo) and the normative meaning (the voice). As mentioned above, an instinctive level useful to work with unknown and new problems, is the iconic one, where the sign and the object share a morphologic resemblance. A group (let us call it A) started with this approach. Here, the drawing stands for the rabbits (Figure 2). Note that a student from group A drew two rabbits of the same color (the problem was about couples of rabbits). The evolution of this naïve approach can lead to two directions, one pursued by group A, another by group B. It is worth noting that both these two groups (A and B) worked graphically to obtain the 13th number of Fibonacci.

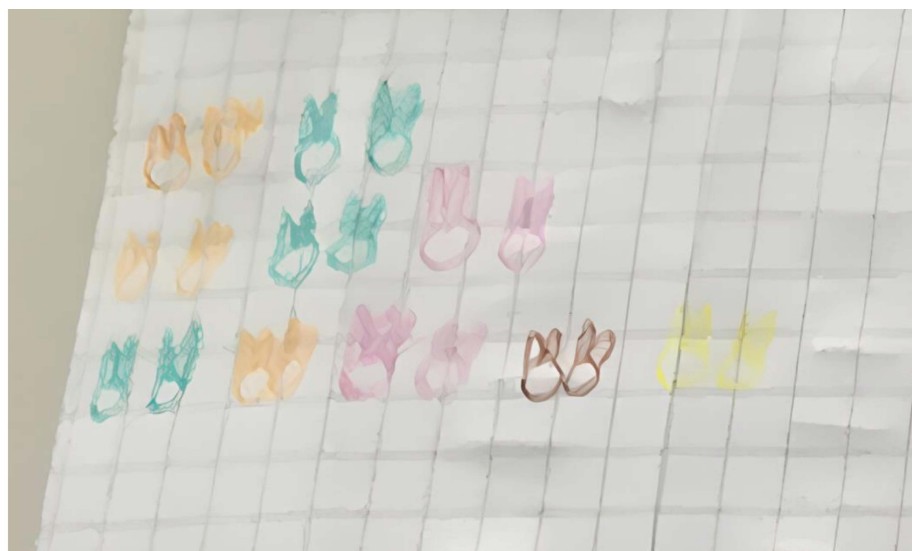

**Figure 2.** Iconic approach to the Fibonacci problem, group A.

Group B started in a more abstract, or symbolic, way, whereby each rabbit is represented by a cross (Figure 3). This is conventional—the relationship between the rabbit (object) and cross (representamen) is not rooted on a physical connection or any kind of similarity. This link is purely conventional, so we can say that this approach is more (but not entirely) symbolic. The two rabbits are still thought of as a pair (a set of two rabbits) and not a couple (a single entity); at the first month, for instance, we have one couple, but group B drew two rabbits, thinking these two rabbits as a pair. For example, this approach led one group to overestimate (double) the number of (couples) of rabbits, while another group reasoned, with more effort, on two levels, the total number of rabbits and, dividing by two, the number of couples (thought of as pairs). Finally, the arrow from one couple to another means that one couple (the left one) generates another one (the right one), making it clear which couple is already fertile. Group C started from the beginning to reason on a symbolic level, representing the rabbit couples as joined circles (oo); these circles are never used separately. The oo symbol resembles two separated entities (rabbits), each denoted as o, so oo seems to stand for "a pair of rabbits"; despite its morphology and origin, it is used, pragmatically, to point out two rabbits at once, so it stands for "a couple of rabbits". It is worth stressing that the difference in the iconic or symbolic aspect is clear to the students themselves; one student specified that, in their group, one sub-group worked on "drawing the rabbits", while the other worked on a "scheme with crosses", or as a group mate stated, with "opposite reasoning". Nevertheless, they all agreed that they obtained a synthesis (question 6 from the questionnaire).

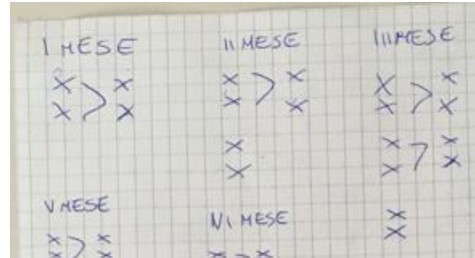

**Figure 3.** From an iconic to a symbolic approach on the Fibonacci problem, group B. "I MESE" means "first month", "II MESE" means "second month", and so on.

In contrast, group A skipped from a purely iconic approach to a more symbolic one. For each month (represented with G, F, etc.) there are lines which represent a couple of rabbits (Figure 4). Note that, in this case, the two rabbits of the first month are represented as a line, so they are thought of as couple. Also note the persistence of color, which is useful to distinguish between the already fertile couples and the younger ones. Again, the lines are not connected in any way with the morphology of rabbits, this indicates a shift towards a completely symbolic approach.

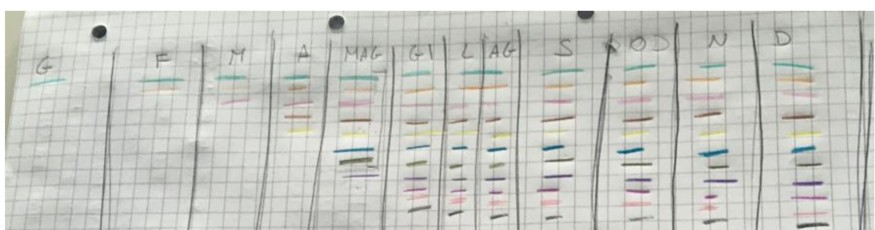

**Figure 4.** Towards a symbolic approach to the Fibonacci problem, group A. "G, F, M, . . ." are the starting letters of the names of the months: "Gennaio", which means January, and so on, until December.

In these contexts, symbolic does not just means arithmetic (although each arithmetic sign is mainly symbolic). The symbolic approach of these groups is based on shared

meanings inside the group; the convention on which the symbol is based is an in-group-convention. Phase b led to a switch from this in-group-*sense* to a class-*meaning*.

This arbitrary aspect of signs and their interpretation, present even in the mathematical language, is clear in the last group (D), where the use of 1 and 0 is employed to indicate the fertility of the couples (Figure 5): a couple is born as a 0, after a month it becomes a 1, so is able to reproduce; from then on, this couple will remain until the end of time as a 1. The strategy employed by this group is minimalistic; the crucial aspect lies in the use of a common sign, even in mathematical practice, which takes on a new aim and, thus, a new meaning (a *sense* in the context of the assigned problem and in-group setting) and a new use. In this context, the syntactical aspects of the mathematical signs are secondary to their pragmatic use, driven by the problems at hand. Therefore, we observe a conflict, requiring a collective discussion to reach a synthesis, between the *sense*, conditioned by the specific problem and group, and the *meaning*, influenced by the mathematical convention and the mathematical inherited knowledge. We will delve further into the group D path because this was a heterogeneous group, which started from graphical approaches, as we can see from Figure 6. In Section 5.4., we will analyze their perception on their evolution, as described in the questionnaire.

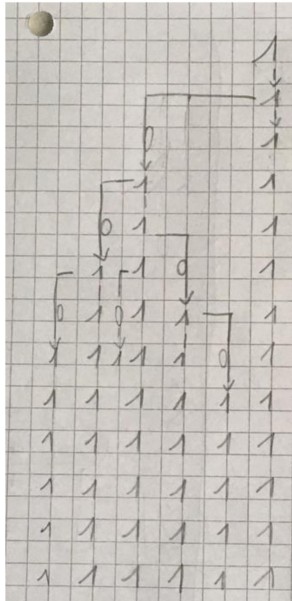

**Figure 5.** The use of mathematics symbols as contextual symbol for the problem, group D.

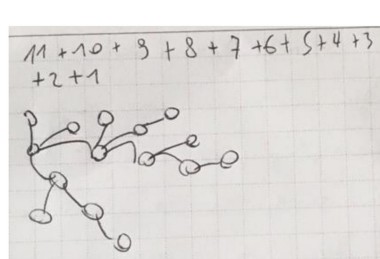 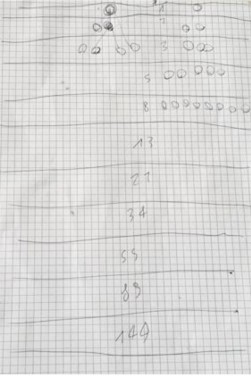

**Figure 6.** The starting points in searching of meanings, group D.

*5.2. Classroom Discussion*

The key point of this section is to emphasize the necessity of a collective discussion (e.g., in the sense of [27], as suggested in [11]) to foster dialogue between the groups and the transition from the *sense*(s) of individual groups to the *meaning*(s) embraced by the entire classroom. This does not imply that each *sense* is vanished/eradicated; rather, they are coordinated and interconnected, representing different paths, stemming from the same problem and leading to different solutions. Sharing senses ultimately leads to meaning, in respect to the pairs (students), to the experts (teacher and researcher), and to the historical problem. These *senses* are the *echoes* to the starting *voice*, Fibonacci's one, and its implicit *meanings*, to interpret. In the next section, we will appreciate the evolution of the groups concerning their interpretation of the given problem.

*5.3. In-Group Problem Posing*

The phase centered on the problem posing or storytelling activity completes the circle; the evolution of in-group senses, empowered by the collective meanings (phase b), materializes in the creation of a new story by each group.

There are several key assumptions Fibonacci made in his problem that the students underline—rabbits never die, become sick, or are poisoned; one couple generates, each time, a couple (they are consistently fertile and they never generate more than one couple or couple of the same gender); there are not genetic issues due the endogamy (based on the students' knowledge of genetics); rabbits never escape or are given away by the breeder; and there are no natural disasters or predators.

Many students answering question 3 expressed how this problem is not close to their reality and it is not a likely scenario. Therefore, during the storytelling phase, they added sentences like "don't consider any trouble". However, this also led to misunderstandings, such as "you know that each couple generates one couple every two month", highlighting how problem posing can reveal the students' implicit difficulties [28]. The voices and echoes game as the discussion are dialectical and endless processes, in line with a Vygotskian perspective. Therefore, the outputs mentioned above represent stages upon which to continue working. We even have a mature re-elaboration (original in Figure 7).

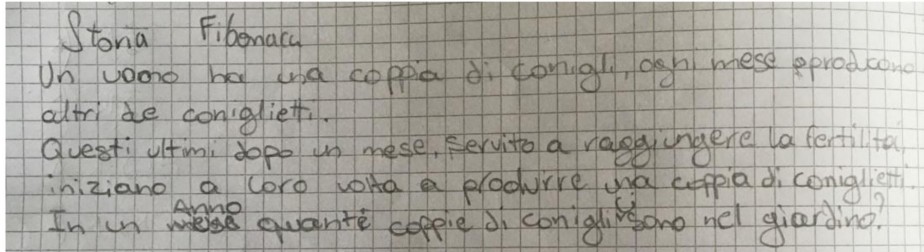

**Figure 7.** The new story on rabbits, inspired by Fibonacci's original one.

> A man has a pair of rabbits, every month they produce two more bunnies. After a month, when they reach fertility, they in turn start to produce a pair of bunnies. In a year, how many pairs of rabbits are there in the garden?

The syntactical structure of the problem is clear; the explanation provided gives a context and a meaning to the mathematical assumptions and pattern ("needed to reach the fertility"); the change in the nouns illustrates the progression of generations (rabbits generate bunnies, the former is the initial couple of rabbits, the latter are two, or a pair of, bunnies). The use of "in turn" (in the original version, "a loro volta") emphasizes the recursive nature of the problem, highlighting the cyclical pattern of reproduction, where each generation gives rise to the next (bunnies become rabbits and generate bunnies); this points directly towards the rise of recursive thinking in the students involved in the problem-posing activity. This interpretation resembles the approach taken by group D (Figure 5), where 0 (infertile couples or bunnies) become 1 (fertile couples or rabbits).

Finally, all the narrative frame of the story remains coherent (there are no narrative fractures in the sense of [29]); unlike the original Fibonacci problem, where the story merely serves as a container for mathematical data and thinking, here, the story acts as a structural element, closely resembling the recursive pattern and thinking.

*5.4. Questionnaire*

In the questionnaire, the first question provides insight into the students' perception of their own work; students talked about tables divided into months, drawings, graphics, sketches (including numbers and words), tree- or "waterfall"- diagrams, or genealogy trees. More importantly, we can follow the progression of student awareness regarding the problem and its mathematical core. Students from group D talked about the in-group strategy; they started working by themselves, after they shared their partial results to arrive at a collective solution. One student (D.1) wrote in response to question 5: "at the beginning, we had different ideas, after we confronted each other trying to *combine* all the ideas and get a solution *together*". Here, we observe the desire to establish a common and shared meaning and overcome the personal sense. Another student (D.2) from group D, in response to question 4, shows an evolution in thinking not only at a semiotic level, but also on the interpersonal front, stressing that the "rewriting of the problem and the factors that can lead to a different result" (problem-posing phase) as a factor to change their point of view. While the first student (D.1) believed that the in-group discussion/work was enough, the second student (D.2), on the other hand, gave significant importance to the explicit articulation of voice and echoes; D.2 also stressed the evolution of their solution (question 4) due the use of different strategies within the group. Another student (D.3) mentioned "*we* started using numbers, *they* started using schemas", illustrating a fruitful dichotomy, as the subsequent harmonic synthesis that resonates throughout the answers of the group D students. This aligns with the symbolic–iconic dichotomy used to analyze the patterns and strategies produced by the students. Moreover, the relationship between numbers and shared meanings (which makes the solution easier to communicate) such as the one between schemas and personal senses (which implies the need of the students to explain how the schema works). Voices and different personal meanings are also present within each group.

Regarding question 2, almost all students were aware of the emergence of a recursive pattern from month 4 to month 6, where there is a necessity to switch to more powerful and general strategies, overcoming graphical approaches, for instance.

## 6. Discussion and Conclusions

Regarding the research questions, it is evident that the narrative setting and the historical problems, if carefully chosen and adapted (by teachers, researchers, or even by aware students), can serve as excellent environments for exploring new concepts (such as Recursive Thinking) and build mathematical competencies (even in the sense of the four Cs [4]). As many students highlighted in their responses to question 1 of the questionnaire, the initial step involves careful and repeated reading of the text. Secondly, the transition to a cross-semiotic perspective should be related to the in-group cooperation. In group D, which showed a mature and polyphonic construction starting from an iconic approach toward a symbolic one, students were aware of the different opinions (senses) and these opinions were not merely subordinated to one perspective ("the" meaning). This open and variegated semiotic treatment enhanced the dialogue between different ways of seeing and thinking the same problem from various viewpoints. The mathematical word problem, situated in a historical context distant from the students' daily experiences, provided elements for students to work with and express their understanding and strategies, even in friction with the different social, cultural, and historical context of Fibonacci.

The movement of the students is not solely towards an efficient or intelligible symbolic (e.g., arithmetic) approach to reach a solution; rather, the key aspect lies in synthesizing the various hints (graphic factors sustain more abstract ones) and making other groups aware of their ways of using the chosen signs. During the collective discussion, students showed care

regarding their conventions, their meaning, and their use; the collective discussion made all the participants aware of the conventional aspects and choices needed to be made explicit to "others" (e.g., "we *represented* couple of rabbits as crosses"). As each student's sense became the group's meaning (Section 5.1), the group's sense became the class' meaning (Section 5.2). Fibonacci's voice, mediated by the teachers and researchers, created the landscape for a room of echoes and of new approaches. Each echo reverberated on other echoes, creating a deeper and shared meaning—a class voice. The meaning, elaborated collectively, in group and in class, "returned" to the group during the problem-posing phase (Section 5.3). Now, the new "voice" is the fruitful connection of the echoes; this voice, embraced by the teacher and researcher (the echoes become the voice), is now internalized by the students [11,12]. This new voice is closer to the interpretations given by the students, and will build the starting point for new echoes; clearly, students are moving into the ZPD (Zone of Proximal Development) [30]. We see in the making, a dialectics between meanings rooted in culture and history and the present-day students' naïve interpretation [8,10], the overlapping of voices and their echoes [11,12], the dialogue between signs and their meanings [9]—mathematics and its meaning are in the making.

Therefore, we can conclude that the mathematical idea behind the problem, recursion, emerged for the students, where there was a clear conflict between the semiotic power of the naïve approach (graphical, for instance) and the necessity to switch to a more abstract and more compact approach; as stated by one student in group A, "*we started* by a graphic method, *then we switched* to counting". This does not mean that before formalization (using algebraic symbols) the idea of recursion was absent (the recursive pattern is crystal clear in Fibonacci even without formalization), but rather that it was still strongly linked with the starting problem and the original context. In the process of abstraction and switching between signs, there is the construction of the underlying mathematical concept. There is not a contraposition between these approaches, but rather a natural flow. The narrative aspects, as discussed in Section 5.3, and the use of speech and natural language, as metalanguages, are indeed cohesive factors that link history, mathematics, and all semiotics aspects used to face and manage the Fibonacci problem. Thus, the story helped students give a meaning to the problem (Section 5.1) and give (a) new meaning to the (new) problem (Section 5.3), after the collective work (Section 5.2).

Starting from a coherent symbolic level, such as using algebraic formalizations from the beginning, could create the illusion, in students, that they understand and hold the idea of recursion, while, in reality, they may only possess a practical proficiency in formally manipulating the symbols conventionally associated with this concept. The construction of the idea goes through the construction of signs (including iconic and indexical aspects) and the discussion of their meaning in dialogue with peers and common knowledge. This is why our focus is on the *construction of Recursive Thinking*, giving the students the opportunity to face recursion, starting from different contexts.

What has been discussed above are the initial steps of an ample work at different school levels, focused on recursive thinking in all its forms (such as Fractals, Fibonacci Numbers, Factorials, etc.). One branch of this work plan, near to what has been discussed in this paper, involves presenting historical problems to different classrooms, in a rhetorical (or narrative) shape and to make students work on the problems, according to [8–12,15], subsequently re-creating these problems in a more easy-to-read language, as if they were writing to a peer. Direct developments result from introducing elaboration and the structuring of recursive thinking directly through the use of high-level programming languages, especially in advanced high school classes. Fibonacci numbers, along with fractals and factorials, could be particularly fruitful topics for introducing, exploring, and building Recursive Thinking in this perspective.

A further next step would be to focus on the Golden Ratio, in relation to the Fibonacci pattern, and, thus, to focus on R.T., on one hand, and to Kepler's work and geometric approaches, on the other, because, as stated by Mario Livio,

Some of the greatest mathematical minds of all ages, from Pythagoras and Euclid in ancient Greece, through the medieval Italian mathematician Leonardo of Pisa and the Renaissance astronomer Johannes Kepler, to present-day scientific figures such as Oxford physicist Roger Penrose, have spent endless hours over this simple ratio and its properties. But the fascination with the Golden Ratio is not confined just to mathematicians. Biologists, artists, musicians, historians, architects, psychologists, and even mystics have pondered and debated the basis of its ubiquity and appeal. In fact, it is probably fair to say that the Golden Ratio has inspired thinkers of all disciplines like no other number in the history of mathematics [31] (p. 6).

**Author Contributions:** Conceptualization, G.B., A.D. and B.N.; methodology, G.B.; software, G.B.; validation, G.B., A.D. and B.N.; formal analysis, G.B.; resources, G.B., A.D. and B.N.; data curation, G.B.; writing—original draft preparation, G.B.; writing—review and editing, A.D. and B.N. All authors have read and agreed to the published version of the manuscript.

**Funding:** The APC was funded by EduSpaces-MultiLab of the Faculty of Education of the Free University of Bozen-Bolzano, Project leader, Federico Corni.

**Institutional Review Board Statement:** The data used in this study were collected and anonymized by the school teachers according to the school policy, and do not allow any conclusions to be drawn about the participating individuals. The procedure complies with the data protection law (GDPR).

**Informed Consent Statement:** Informed consent was obtained from all the subjects involved in the study.

**Data Availability Statement:** The data presented in this study are available on request from the corresponding author. The data are not publicly available due to privacy.

**Acknowledgments:** This paper resulted from a collaboration starting during the BrEW Math 01 (Brixen Education Workshop on Storytelling in STEM disciplines at the crossroads of science and humanities) held at the MultiLab of the Faculty of Education, Free University of Bozen-Bolzano, 8–10 November 2022. We would like to thank the Elisa and her marvellous class 2MGC of Istituto Aldini Valeriani, Bologna (Italy), who made this exploratory study possible and contributed with rich and precious feedback, suggestions and passion.

**Conflicts of Interest:** The authors declare no conflict of interest.

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
