# Peer review of "Fibonacci Numbers between History, Semiotics, and Storytelling: The Birth of Recursive Thinking"

_education, doi:10.3390/educsci14040394_

Round 1

Reviewer 1 Report

Comments and Suggestions for Authors

The basic idea of the paper is well presented, documented, and it offers a clear description of the method which has been used. The presentation forms, the analyse of data, the discussion and conclusions are coherent with the subject chosen. It underlines that a good combination of a narrative setting and historical problems, if adapted and carefully chosen, by teachers/researchers, can be a great environment to explore new concepts and to build mathematics competences.

Reviewer 2 Report

Comments and Suggestions for Authors

The paper "Fibonacci Numbers between history, semiotics, and storytelling: the birth of recursive thinking" integrates history, storytelling, and semiotics in an experiment in mathematics education through the lens of Fibonacci's rabbit problem offering a novel approach in teaching recursive thinking.

The work is well-organized, starting from historical backgrounds to theoretical frameworks, and leading to practical classroom application. It comprehensively describes the methodology, analysis, and findings. While the paper, generally speaking, appears scientifically sound and grounded in established educational theories and methodologies, there is a part dealing with the relations between mathematics education and history of mathematics, in a socio-cultural perspective, that, in my opinion, needs to be clarified (see below).

Regarding the use of English, although the overall meaning of what is written can be quite well understood, there are some sentences that are wordy or unclear, and others that are very difficult to interpret. For example, I wasn't able to make sense of the following sentence in line 402: "We will deepen the D group parabola". Does it mean "we will delve into the parabola-shaped graph made by group D"?

The article could definitely benefit from a review by a native speaker.

 The manuscript will be suitable for publication after the following comments are addressed.

From line 288 onwards, the problem of the relationship between mathematics education and history of mathematics begins to be addressed. It is said that the problem should be interpreted ("decrypted") by the students according to Fibonacci's context (his time, his aims, etc.). But, while there is, indeed, a distance between the problem, as formulated, and the students' daily life and experience, it is more a distance due to geography than history, so to speak. In other words, the rabbit problem, as it is formulated, would not be so far removed from a student who lives in the countryside. On the other hand, we are dealing here with the concept of recursion. And this indeed presents epistemological peculiarities that not only can be difficult for students to grasp but also were not clear as such in Fibonacci's time.

I think that an additional phrase here is in order to clarify this issue, precisely to avoid the danger, that Redford himself highlights in the article you cited, of seeing past mathematical achievements as clumsy effort that tended towards the modern conceptual formulation.

Speaking about style and formatting, it is advisable to avoid the use of contractions (like: it's, don't, etc.) in a formal register: always use full forms (it is, do not, etc.). It is also  advisable to follow the standard typographical practice of using italics for algebraic  variables and using the minus sign (−), instead of the hyphen (-), to indicate subtraction. To avoid gender-exclusion issues, it is also preferable to write generic pronouns in the plural form, i.e. they, instead of his/her.

There are several typos and poorly chosen terms in the paper that need to be reviewed. Below is a non-exhaustive list, the first number indicated is the line number.

6 maybe "form" is a better term than "shape", in this context.

9 "recursion", instead of "the recursion"

22 "Plato", instead of "Platon"

24 "The Assayer", instead of "The Assaye"

49 better "circa 1170 – circa 1240/1250"

54 "treat", instead of "treats"

55 "chapter 12", instead of "the chapter 12"

58 "as follows", instead of "as following"

113 "intersects", instead of "intersect"

127 "delve deeper into", instead of "deepen"

134 "the aim", instead of "aim"

137 "educational aims", instead of "education aim"

141 "useful for", instead of "useful to"

169 "everyday and scientific" instead of "the every day and the scientific"

180 "scientific concepts", instead of "the scientific concepts"

188 "most", instead of "more"

193 "to", instead of "with"

220 "to work", instead of "work"

221 "History", instead of "the History"

233 "a historical", instead of "an historical"

241 "the whole in-classroom activity revolves", instead of "all the in-classroom-activity goes"

246 "for a", instead of "to a"

247 "of each", instead of "for each"

250 "classic" (or even better, "typical"), instead of "classical"

256 "except" instead of "expect"

264 "northeastern", instead of "notherneastern"

284 "with", instead of "into"

286 "trial", instead of "try"

314 "description/story", instead of "description/history"

321 "limited", instead of "bounded"

322 "due to", instead of "due"

354 better "pursued" than "travelled"

361 "Group B", instead of "The group B"

391 "signs", instead of "sign"

391 "mathematical", instead of "the mathematics"

392 "is", instead of "it's"

453 "In", instead of "Into"

527 "forms", instead of "shapes"

528 "what is discussed", instead of "what discussed"

533 "geometric", instead of "geometrics"

Comments on the Quality of English Language

The manuscript will be suitable for publication 

Reviewer 3 Report

Comments and Suggestions for Authors

The paper represents the contribution to the corpus of knowledge primarily in the field of education science, but it also has a significant value in an applicative sense in the context of application in the field of communication science and a practical contribution to high education. The idea is clearly described. Research methods, results and conclusions are very well connected. The topic of the paper is clearly related to the Special Issue: Storytelling in STEM Disciplines—At the Crossroads of Science and Humanities. The research questions are very well elaborated. The idea is clearly described with defined methods, results and conclusions. The paper is well logically structured. The sessions follow the logic of presenting the themes described in the abstract well and their excerpts clearly describe the (subject) session. Paper contains an original approach to research.

Remarks/Suggestions:

(1)   Abstract: The idea is described, but without clearly defined methods and result, conclusion and recommendation for further research. Suggestion - correct the summary, e.g. State the purpose of the research very clearly in the first sentence… Describe the methodology in the second sentence… Report the major findings of the research in the third sentence… Give your interpretation of the impact of the research in the fourth sentence. Tell the reader whether the research will or should produce a change in scientific thinking or practice. In particular, what gap in knowledge has the research filled?... Count words. An effective abstract should be about 200 words. Suggestion - expand the abstract.

(2)   Format: The authors must revise the paper to exactly match the MDPI format (Use the Microsoft Word template to revise your paper, https://www.mdpi.com/journal/education/instructions#preparation). The paper should have a form: 1 Introduction, 2 Materials and Methods, 3 Results, 4 Discussion, 5 Conclusions (Conclusions is not mandatory but can be added to the manuscript if the discussion is unusually long or complex).

(3)   Discussion & Conclusions: Suggestion - Reformulate the chapter Conclusion into Discussion + Conclusion (in which you will present only the main conclusions you have reached).

(4)   Figures: Figures are technically well executed but not linked with the main text. Suggestion – linked/cited them the main text as Figure 1, Figure 2, etc.

(5)   Further research: The paper does not contain guidelines or recommendations for further research. Suggestion - describe in the "Conclusion" thoughts on the possible implementation and continuation of the research and/or add a short recommendation for further research in the abstract.

(6)   References: The references are adequate in terms of content, but insufficient in the field of storytelling. Authors have only 2 of the 27 references from the last 5 years. Each reference relates/linked to the main text. Suggestion - reinforce them using publications published in the past 5 in the relevant databases (WoS, Scopus).

(7)   List of References:  The authors must revise the list of references to exactly match the MDPI format.

(8)   Author Contributions: Suggestion - Contribution of Individual Authors to the Creation of a Scientific Article.

I do encourage the authors to carry on research work, expand their research questions and apply new analysis methods.

Round 2

Reviewer 3 Report

Comments and Suggestions for Authors

I do encourage the authors to carry on research work, expand their research questions and apply new analysis methods.